# EDIT-BENCH: EVALUATING LLM ABILITIES TO PERFORM REAL-WORLD INSTRUCTED CODE EDITS

**Wayne Chi**[1,*]    **Valerie Chen**[1,*]    **Ryan Shar**[1,*]
**Aditya Mittal**[1]    **Jenny Liang**[1]    **Wei-Lin Chiang**[2]    **Anastasios Nikolas Angelopoulos**[2]
**Ion Stoica**[2]    **Graham Neubig**[1]    **Ameet Talwalkar**[1,†]    **Chris Donahue**[1,†]

[1]Carnegie Mellon University    [2]UC Berkeley, Arena

## ABSTRACT

Instructed code editing, where LLMs directly modify a developer's existing code based on a user instruction, is becoming a widely used interaction mode in AI coding assistants. However, few benchmarks directly evaluate this capability and current datasets often rely on artificial sources. We introduce `EDIT-Bench`, a benchmark for evaluating LLM code editing capabilities grounded in real-world usage, i.e., user instructions and code contexts collected in the wild. `EDIT-Bench` comprises of 540 problems, multiple natural and programming languages, and a diverse set of real-world use cases, ranging from resolving errors to adding features. `EDIT-Bench` introduces context-dependent problems that require the model to understand code context, highlighted code, and cursor position in addition to the user instruction. We evaluate 40 diverse LLMs and observe that `EDIT-Bench` is a challenging set of problems where only 1 model scores over 60%. We find that model performance varies across different categories of user instructions. Further, we find that different levels of contextual information greatly affect task success rate, with performance varying up to 8%, indicating the importance of evaluating with realistic context.

    🔾 **Github Repo**     `https://github.com/waynchi/editbench`
    ⛬ **Leaderboard**     `https://waynechi.com/edit-bench/`

## 1 INTRODUCTION

Software developers increasingly write code with AI assistants such as Github Copilot (Github, 2022), Cursor (Cursor, 2023), and Continue (Continue Dev, 2025) using a variety of modes of interaction. *Instructed code editing*, where developers use natural language to request the assistant to edit a highlighted section of code, has emerged as a prominent interaction mode alongside autocomplete suggestions and chat (Nam et al., 2025). Due to the flexibility provided through natural language instructions, use cases for edits are diverse and range from code improvements given detailed user instructions to bug fixes provided only an error trace (Cassano et al., 2023b). Because of this, instructed code edits pose a challenging set of problems that existing LLMs must tackle to support developers.

Despite the emergence of this new interaction modality, we lack benchmarks to capture real-world edit behavior. Code generation benchmarks typically evaluate LLM capabilities on generating code from scratch (Chen et al., 2021; Austin et al., 2021; Jain et al., 2024; White et al., 2024). While there are a few edit-related datasets (e.g., CanItEdit (Cassano et al., 2023b), Aider polyglot (Gauthier, 2025)), the sources of data are not reflective of most real-world software development, relying on either simple, annotator-written problems or Leetcode and educational style problems that do not capture diverse, real-world software development challenges. Recent work has begun collecting human preferences to interactively evaluate models—Chatbot Arena (Chiang et al., 2024) evaluates LLM capabilities for chat and contains a coding subset, while Copilot Arena (Chi et al., 2025)

---
    * Equal contribution.
    † Equal senior author.

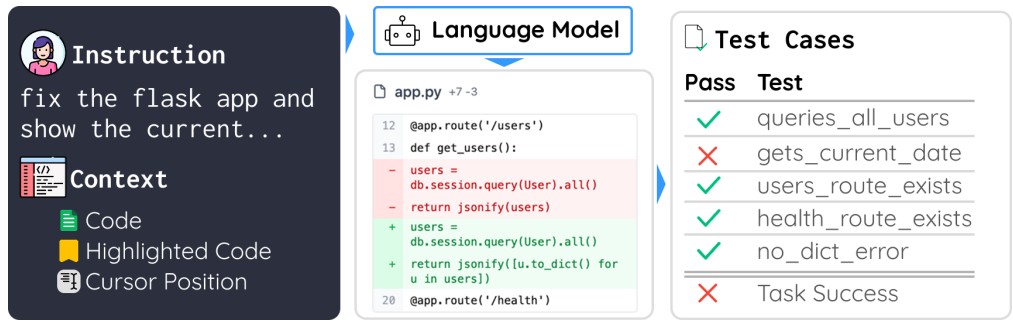

Figure 1: **EDIT-Bench tests LLMs' real-world editing capabilities.** We propose EDIT-Bench, an evaluation on real user instructions and code snippets collected in-the-wild. It is the first benchmark for instructed code edits that requires models to ingest the user instruction, current code, highlighted code, and cursor position to solve problems.

evaluates LLM capabilities to perform code completions—highlighting a growing awareness of the need for grounding evaluations with in-the-wild data. However, "arena-style" evaluations are costly, requiring a significant number of human votes to rank a new model.

We introduce EDIT-Bench, a benchmark for evaluating LLM code editing capabilities built on real-world edit contexts and instructions (Figure 1). We source our problems by developing a VS Code extension that mimics existing instructed code editing tools from GitHub Copilot and Cursor. As developers use the extension, we gather a live, in-the-wild dataset containing user-written instructions, associated code context, and user votes between pairs of model responses. We recruited nearly 500 users to provide these data points. EDIT-Bench differentiates from previous edit-related benchmarks in several ways:

**Diverse user instructions and context.** Since EDIT-Bench is constructed from data collected from programmers performing day-to-day coding tasks, users specify user goals with diverse content and formats. For example, a bug fix can be requested as "fix this" accompanied with highlighted code, a direct dump of the error trace, or a natural language description of the erroneous behavior. EDIT-Bench tests for these varied user instructions instead of the more templated approaches (e.g., fix a specific function in a well-defined way) in previous benchmarks.

**Context dependent problems.** Real instructed code edits often feature ambiguous user instructions that require contextual clues to parse the underlying user intent. In addition to the user instruction, in EDIT-Bench we also capture the code file to edit, the highlighted region of code, and the user's current cursor position. Code context length can be significant (e.g., $\geq$10k characters), requiring the model to properly use the comments, highlighted code, and other contextual clues to determine the correct solution. We are the first benchmark to include this combination of features for instructed code edits.

**Multiple natural and programming languages.** While most previous coding benchmarks consist of only English problems, EDIT-Bench consists of 5 natural languages (English, Spanish, Russian, Chinese, Portuguese) and 2 programming languages (Python and Javascript). Since our code is gathered in-the-wild, any natural language variations occur in both the user instruction and code itself.

We evaluate 40 open-weight and closed models on EDIT-Bench and find that the best model, claude-sonnet-4 (Anthropic, 2023), achieves a pass@1 of 66.67%. Closed-source models tend to outperform open-weight models, with deepseek-chat-v3.1 and kimi-k2-0905 being the only two open-weight models in the top 10. We observe that both the inclusion of additional context (e.g., highlighted code and cursor position) and the type of edit category (e.g., optimization versus bug fixing tasks) drastically affects performance, Finally, we find that EDIT-Bench is only weakly correlated with existing edit benchmarks like Aider Polyglot (Gauthier, 2025), suggesting that our real-world data captures a unique set of difficult edit tasks. Our results show that EDIT-Bench is challenging even for state-of-the-art models and reveals new insights into model capabilities, emphasizing the importance of benchmarking LLMs on realistic data.

Figure 2: We develop an open-source VSCode extension to collect real-world edits.

## 2 RELATED WORK

**Coding Benchmarks.** Static benchmarks, e.g., HumanEval (Chen et al., 2021) and MBPP (Austin et al., 2021), largely focusing on interview-style programming problems have been the most commonly used to evaluate coding capabilities (Lu et al., 2021; Nijkamp et al., 2023; Zhu et al., 2022; Wang et al., 2023; Liu et al., 2023; Jimenez et al., 2023b; Khan et al., 2023; Yan et al., 2023; Cassano et al., 2023a; Muennighoff et al., 2023; Dinh et al., 2023; Yang et al., 2024b), measured using `pass@k`. Additionally, some recent work focuses on creating live benchmarks that reduce contamination risks (Jain et al., 2024; White et al., 2024). Increasingly, people are interested in code editing with LLMs, focusing on bug fixing (Zhang et al., 2023b; Moon et al., 2023; Shinn et al., 2023; Chen et al., 2023; Olausson et al., 2023; Jin et al., 2023; Joshi et al., 2023; Wei et al., 2023; Li et al., 2022), a specific subset of code editing; fill-in-the-middle code completion (Bavarian et al., 2022; Fried et al., 2023; Yee & Guha, 2023; Roziere et al., 2023; Guo et al., 2024a; Zhang et al., 2023a), an inference strategy that requires specific insert locations; and intrinsic code editing (Li et al., 2023; Gupta et al., 2023), which involves editing code without a specified instruction, exerting the model's ability to intrinsically ascertain the desired code changes. CodeEditorBench (Guo et al., 2024b) evaluates code editing using competitive programming problems and CanItEdit (Cassano et al., 2023b) expands on this to create varied prompts and diverse topics.

**Grounding Evaluation in Real-World Data.** A limitation of the aforementioned benchmarks is that the source of their tasks is not from real-world user data. Copilot Arena (Chi et al., 2025) evaluates code completions with real-world data and highlights how the distribution of data from benchmarks differs from real-world data in terms of the type of task, context length, and more. However, these in-the-wild evaluations require immense scale to build a leaderboard and evaluate new models (e.g., Chatbot Arena (Chiang et al., 2024) has millions of votes). The primary benchmark that creates problems from real-world sources is SWE-Bench (Jimenez et al., 2023a) and related extensions including SWE-Bench Multimodal (Yang et al., 2024a) and Multi-SWE-Bench (Zan et al., 2025). However, these benchmarks focus on fixing issues that require agentic workflows (e.g., editing multiple files) and are limited to a handful of repositories or problems written in one natural language. Our work, `EDIT-Bench`, complements this growing set of benchmarks by providing a benchmark for instructed code edits that is *realistic* (i.e., collected from real users in real workflows) and *diverse* (i.e., contains many different natural languages and task categories).

## 3 BENCHMARK CONSTRUCTION

### 3.1 DATA COLLECTION.

We develop an open-source VSCode extension with instructed code editing as a core feature to support the collection of code edit data. Gathering data via a real coding extension (Izadi et al., 2024; Chi et al., 2025) allows for more realistic instructions and tasks when compared to coding competition platforms. For each code edit, the user highlights a code-snippet and writes a short task description (Figure 2). Participants are not compensated for using the extension, as in a traditional user study, but instead receive free access to state-of-the-art models. Given the sensitive nature of

programming, we established clear privacy controls to give users the ability to restrict our access to their data. Depending on privacy settings, we collect the user's instruction, code context (including the highlighted code segment, the cursor location, prefix, and suffix) at the time of the request, and model responses. Additionally, we log whether the user accepted the edit. Our data collection process was reviewed and approved by our institution's IRB. Additional details about our data collection policy are provided in Appendix A.

## 3.2 PROBLEM CURATION.

Across 458 users, we collected 2672 responses (i.e., the user accepted an edit). However, not all of these responses were interesting, challenging, or even feasible to turn into testable problems. We narrow our problem set in the following ways. First, we focus on questions written in Python and Javascript, which combined comprise of the majority of our responses at just over 1700 problems. Second, we exclude problems that are too similar to one another—sometimes a user might try similar prompts on the same code context to see how different models edit. Lastly, we remove any trivial (e.g., add a single parameter), stylistic (e.g., add a comment), or ambiguous problems. We provide concrete examples of removed problems in Appendix C. This filtering process left us with around 470 problems which we found both interesting and challenging. Given that not all problems are feasible to create test harnesses for, we succeeded in creating 109 unique problems for `EDIT-Bench-core`. There are five languages—English, Russian, Chinese, Polish, and Spanish —in `EDIT-Bench`. In order to equally distribute the natural languages in the problem set, we also translate each problem to the other languages found in our problem set to form `EDIT-Bench-complete`. To do so, we followed a similar method prescribed by HumanEval-XL (Peng et al., 2024) and translate the comments in each problem using GPT-4o to create a total of 540 problems. To validate the translations, we had native speakers evaluate a subset of the translated tasks, primarily in Chinese and Spanish. In addition to GPT-4o, we experimented with several other models (GPT-4o-nano, GPT-4o-mini) and Google Translate, but found GPT-4o to provide the best quality with no noticeable concerns with any of the translations.

## 3.3 TEST HARNESS CREATION.

The data from our extension provides us with realistic human instructions and code, but does not contain test cases, making the raw data ill-suited for a benchmark. We create test harnesses composed of the *environment setup*, which includes preparing configurations, virtual environments, or mock files, and *test cases* that define expected inputs and outputs.

To write our tests, we assemble a team of five experienced programmers who have expertise in both natural and programming languages present in the real-world edit data. The team, recruited through academic networks, included researchers and students from various fields who write code extensively. The annotators were instructed to create test harnesses that adhere to the user's intent and are generalizable to different potential implementations. While the user instruction and code file are perhaps the most important pieces of information, they by themselves can often be too ambiguous. The highlighted code segment and cursor locations provide crucial contextual clues to prescribe user intent. Annotators were asked to design problems given all of this information, and if a problem was still too ambiguous, we asked the annotators to remove the problem. To support the annotation process, we generated some example solutions using GPT-4o and Sonnet 3.7 (chosen to balance cost and quality) to give insight into possible solutions. Additionally, annotators were also asked to screen for and remove any Personal Identifiable Information (PII). Finally, all refined test cases were assigned to a second annotator in the team to do a second review with the same procedure.

Originally, we attempted to use a coding agent (e.g., Claude Code) to construct test cases, but found that the agent often struggled with test case generation itself, frequently resorting to undesirable tests such as directly pattern-matching with the source code, despite explicit instructions to avoid this behavior. However, despite the complexities involved in environment setup, especially for languages such as Javascript, we found the agent was consistently able to set up the correct packages and environments. As a result, we used the agent to setup the test harness environment. We provided setup files (e.g., a `conftest.py` file in Python and a `jest-config.js` file for Javascript) to help support the agent and standardize outputs.

Table 1: **Comparing `EDIT-Bench` to other edit-related benchmarks.** We compare `EDIT-Bench` with similar benchmarks (CanItEdit (Cassano et al., 2023b), EditEval (Hu et al., 2023), Aider Polyglot) in terms of the problem source, user instruction (# NL refers to the number of natural languages), code context (# PL refers to the number of programming languages, HL refers to whether users can highlight a subset of code), and associated test cases. Standard deviation is indicated by $\pm$. `EDIT-Bench` is the only benchmark built from in-the-wild problems and exhibits considerable variation in both instruction and code context length.

| Benchmark | Problem | | Instruction | | Code Context | | |
|---|---|---|---|---|---|---|---|
| | # Problems | Source | # NL | Length | # PL | Length | HL |
| CanItEdit (Cassano et al., 2023b) | 105 | Annotator | 1 | $140 \pm 105$ | 3 | $1309 \pm 1116$ | No |
| EditEval (Hu et al., 2023) | 194 | Annotator | 1 | $99.9 \pm 49.3$ | 1 | $258 \pm 185$ | No |
| Aider Polyglot (Gauthier, 2025) | 225 | Coding Exercises | 1 | $606 \pm 885$ | 5 | $6184 \pm 6452$ | No |
| **`EDIT-Bench`** | 540 | In-the-wild | 5 | $238 \pm 738$ | 2 | $5642 \pm 7567$ | Yes |

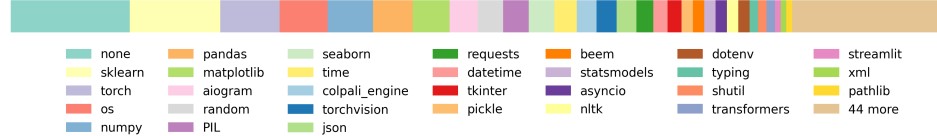

Figure 3: **Distribution of libraries in `EDIT-Bench` for Python problems.** `EDIT-Bench` contains 74 unique imports compared to 25 (CanItEdit), 15 (Polyglot), and 16 (EditEval) from other benchmarks. See Appendix C for other languages and other benchmarks.

# 4 BENCHMARK STATISTICS

`EDIT-Bench` consists of 540 problems that span 5 natural languages (English, Spanish, Russian, Chinese, Portuguese) and 2 programming languages (Python and Javascript). `EDIT-Bench` features a diverse set of problems with considerable variation in instruction and code context lengths (Table 1). Based on the import library usage (Figure 3), we can see that `EDIT-Bench` captures 74 different unique imports, demonstrating much more diversity (at least three times) than existing benchmarks. From our analysis on `EDIT-Bench` problems, we find the following characteristics:

**Real user instructions are diverse and messy.** When inspecting real-world data, we find that users write varied instructions across many problem categories. While many of these categories are similar to existing benchmarks, we find that user instructions are much more informal and less well-specified compared to the annotator-written instructions in existing benchmarks (Table 5). Interestingly, even the way a user would write an instruction within a category varies in terms of descriptiveness. For example, to resolve errors, users may briefly describe the erroneous behavior using natural language or directly paste in the terminal error traces. Further, unlike prior benchmarks where user instructions are only written in English, we find users write instructions in multiple languages, including Russian, Chinese, and Spanish (see Table 1 for additional comparison of user instructions).

**Real-world code contexts span many applications and context lengths.** We observe that users work on a variety of applications, including frontend/backend, machine learning, and algorithmic problems. Additionally, the context lengths are much longer than those evaluated in prior benchmarks (Table 12). We also look at the distribution of code-to-edit token lengths, as computed by the number of highlighted tokens, and find that most people are highlighting targeted portions of code for edits. The median is 138 tokens, while the full file is typically closer to 4.5k tokens. The code contexts that we collect are primarily in Python (43%), with the next most common programming languages being Javascript/Typescript (21%), PHP (18%), and HTML (7%). We focus on problems written in Python and Javascript, which together comprise the majority of in-the-wild instructed edits collected.

**We identify four common clusters of functional edits.** By analyzing in-the-wild user instructions in `EDIT-Bench`, we derive four different categories that describe functional real-world edits: *feature addition*, *feature modification*, *bug fixing*, and *optimization*. We find the distribution across these categories as 43% additions, 27% modifications, 22% fixes, and 8% optimizations. Table 2 provides examples of each category. In our later analysis, we compare how well models are able to perform these different problem categories.

Table 2: **Comparing user instructions written in IDE to the instructions written by human annotators.** We provide examples across different task categories, comparing with two edit-related datasets (CanItEdit (Cassano et al., 2023b) and EditEval (Hu et al., 2023)). We truncate some instructions for brevity and provide full examples in Appendix B. In general, we find that real-world prompts are much less specified and require models to leverage the provided context, compared to existing benchmark prompts.

| `EDIT-Bench` | CanItEdit (Cassano et al., 2023b) | EditEval (Hu et al., 2023) |
|---|---|---|
| **Feature Addition** | | |
| `take the globe countries layer from below ``// this'' and add it to the existing globe` | `Add a method 'estimate_location' that returns the estimated the appropriate location for this house, calculated by...` | `Add a function 'filter_odd_numbers' to filter odd numbers using lambda function.` |
| **Feature Modification** | | |
| `do not use R style, use python style` | `Flip the correlation function given to calculate the covariance instead using the Corr(X, Y), Var(X) and Var(Y). The new function should...` | `Modify the function to correctly determine the season based on month and day, considering edge cases for season changes. Raise error when...` |
| **Resolve Errors** | | |
| `RuntimeError: Cannot close a running event loop sys:1: RuntimeWarning: coroutine 'Application.shutdown' was never...` | `Fix combination_unlimited _rep() so that it returns the right result. The function combination _unlimited_rep should...` | `Fix the bug in 'sum_even_and_even_index' to make it return the sum of even numbers at even indices.` |
| **Optimize Code** | | |
| `optimize the computation by better batching the latter part` | `Optimize the bm25 algorithm by avoiding frequency calculations.` | `Optimize the function to find the longest common subsequence for the given two sequences using dynamic programming` |

## 5 EVALUATION

We now use `EDIT-Bench` to evaluate models and identify trends in code editing capabilities across models. We also compare `EDIT-Bench` results to existing benchmarks. We overview our choice of LLMs, evaluation metrics, and prompts to perform code edits, with additional details in Appendix D.

**Model choices.** We select 40 LLM spanning multiple model families, sizes, and training schemes (e.g., reasoning and non-reasoning models). We use 9 models from the GPT family (OpenAI, 2025), 8 models from Qwen (Hui et al., 2024), 5 models from Llama (Meta, 2025), 4 models from Mistral (Mistral, 2025), 3 models from Sonnet (Anthropic, 2023), 3 models from Gemma (Team, 2025b), 2 models from Grok (Grok, 2025), 2 models from Deepseek (DeepSeek-AI et al., 2024), 2 models from Gemini (Google DeepMind, 2025), 1 model from Kimi (Team, 2025c), and 1 model from the GLM family (Team, 2025a). For a full list of models, see Table 6. For GPT reasoning models (`gpt-o3-mini`, `gpt-o4-mini`, `gpt-5`), we also vary reasoning effort. We set temperature to 0 when possible to reduce non-deterministic outputs.

**Evaluation Metrics.** Following prior work (Kulal et al., 2019; Chen et al., 2021), we report `pass@1`, where 1 code sample is generated per problem and a problem is considered solved if it passes all unit tests. To facilitate analysis on the types of problems that current models excel or struggle with, we also partitioned our dataset into two subsets of `easy` and `hard` difficulty, in addition to reporting the `Full` results. We categorized problems that were solved by $k$ or fewer models as `Hard` and the remainder as `Easy` (Gauthier, 2025). To obtain a roughly even split between problems, we selected $k = 20$. We find that `easy` versus `hard` problems are roughly evenly distributed across problem categories.

**Code Editing Methods.** In all our prompts, the model is given the user instruction and main code context and requested to edit the entire file by regenerating the entire code context. We also evaluate models when given varying levels of contextual information (e.g., highlighted code and cursor position). We find that models perform best when given highlighted code, but not cursor position;

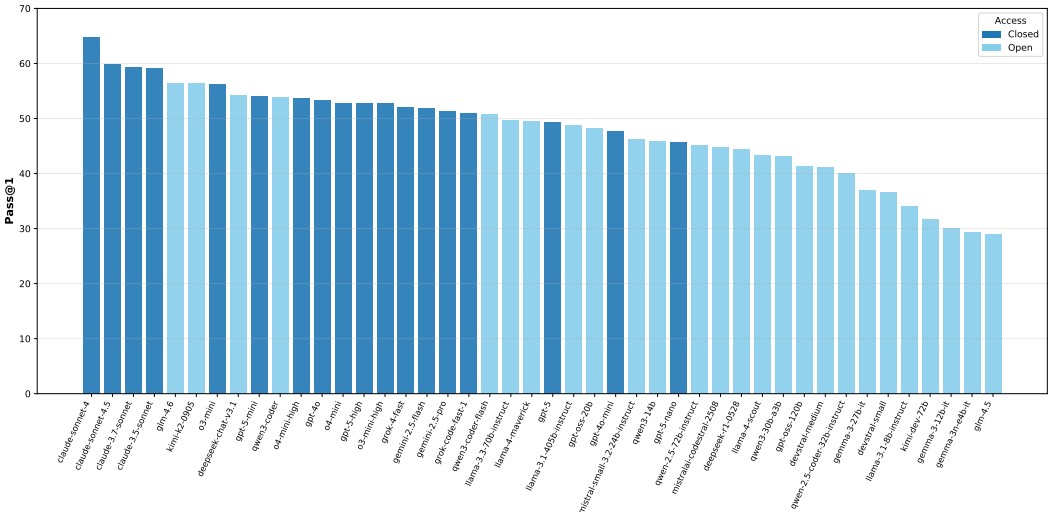

Figure 4: **We evaluate 40 LLMs on `EDIT-Bench`.** We report the `pass@1` of each model; only 1 out of 40 models have a `pass@1` greater than 60%. In general, closed-source models outperform open models.

hence, we run all of our main experiments with highlighted code given only. All prompts are provided in Appendix D.

## 5.1 DISCUSSION OF RESULTS

We present our primary results in Figure 4 and highlight the key takeaways below. Appendix E provides additional results and discussions.

**`EDIT-Bench` is a challenging benchmark, even for current state-of-the-art models.** Only 1 out of 40 models (`claude-sonnet-4`) achieves more than a 60% `pass@1`. Further, EDIT-Bench captures questions of varying difficulty, reflecting the diversity of challenges in real-world code edits. As such, we find a sharp contrast between the `easy` and `hard` questions, where the average gap across models is 59.3% (standard deviation of 10.6%). Given the large gap between `easy` and `hard` problems, we explore what types of prompts are present in `hard` problems compared to the general dataset. Overall, we see that `hard` instructions tend to have *shorter* instructions (by nearly 5 times) but slightly *longer* highlighted code. This means that the model cannot simply rely on following the user's instructions alone but rather needs to reason about multiple pieces of information. We provide an example in Appendix E.

Table 3: **Additional context affects performance.** Highlighted code is crucial to performance, improving task success rate across 5 out 7 models when included in the prompt. Surprisingly, adding cursor position leads to mixed results. Models chosen are the best model in the top 7 model families.

| Model Name | Pass@1 | | | |
|---|---|---|---|---|
| | **Code Only** | **+Highlight** | **+Cursor** | **+Highlight +Cursor** |
| `claude-sonnet-4` | 62.41 | 64.81 (+2.40) | 63.15 (+0.74) | 64.26 (+1.85) |
| `deepseek-chat-v3.1` | 51.48 | 54.26 (+2.78) | 53.15 (+1.67) | 52.78 (+1.30) |
| `gemini-2.5-flash` | 52.59 | 52.96 (+0.37) | 52.41 (-0.18) | 56.30 (+3.71) |
| `kimi-k2-0905` | 54.63 | 56.48 (+1.85) | 52.22 (-2.41) | 58.15 (+3.52) |
| `glm-4.6` | 52.96 | 56.48 (+3.52) | 52.22 (-0.74) | 44.81 (-8.15) |
| `o3-mini` | 60.00 | 56.85 (-3.15) | 59.26 (-0.74) | 55.19 (-4.81) |
| `qwen3-coder` | 56.48 | 53.89 (-2.59) | 56.48 (+0.00) | 53.89 (-2.59) |

**Model performance is affected by additional contextual information.** To evaluate how additional contextual information (highlighted code and cursor position) affects model performance, we run an ablation with the 7 top models in different model families (Table 3). When adding highlighted

Figure 5: **Comparing top-performing open-weight and closed models.** To illustrate individual LLM differences, we compare 7 models and find `pass@1` varies greatly depending on the problem category. Additionally, different models perform best at different categories.

code to the prompt, the task success rate increases for 5 out of the 7 models. On the other hand, additionally adding the cursor position leads to mixed performance when compared to only adding the highlighted code. We notice that trends are generally consistent; the two models that do not benefit from including highlighted code in context—`o3-mini` and `qwen3-coder`—do not benefit from including cursor position either. These findings show the importance of evaluating models on editing tasks that require integrating multiple pieces of information.

**Gap between closed and open models.** Comparing the colors in Figure 4 very readily shows that open models tend to lag behind closed models. Out of the 40 models we evaluate, only 4 out of the top 15 are open models, and the bottom 15 are all open models. Of the open models, we find that `glm-4.6` performs the best with a `pass@1` of 56.48%, with `kimi-k2` and `deepseek-chat-v3.1` not far behind. Surprisingly, `gpt-5` with default reasoning (medium effort) lags behind `gpt-5-mini`. When inspecting test cases where `gpt-5` failed, we find that it struggles with simple tasks like formatting code indentation properly and catching edge cases, despite being a strong reasoning model.

**Models excel in different problem categories.** When we divide questions into categories that test different editing-related skills, we find that performance varies. Overall, we find that models perform best on bug fixing problems (average of 52.2%), which may be most akin to tasks found in prior benchmarks like SWE-Bench (Jimenez et al., 2023a). In contrast, models tend to struggle with optimization and feature addition (44.6% and 39.6%, respectively). Still, we find that `claude-sonnet-4` ranks first in every category except optimization. Furthermore, we find that some models have particularly large gaps between categories (Figure 5). For example, `qwen3-coder-flash`'s top category is fixing bugs while `claude-sonnet-4`'s is making feature modifications.

## 5.2 COMPARISON TO EXISTING BENCHMARKS

We compare our results with two maintained leaderboards: performance on Aider Polyglot (Gauthier, 2025), which has been used in prior model releases as a metric of model editing capabilities, and ranking on the coding subset of Chatbot Arena (Chiang et al., 2024), which has been widely used to capture human preferences. We have 17 and 30 shared models, respectively. We observe a weak, positive correlation with both Polyglot (Pearson correlation coefficient $r = 0.24$, $p = 0.06$) and Chatbot Arena ($r = 0.11$, $p = 0.01$).

We believe our observations are due to the following factors. The first is **code-centric input and output.** Input/outputs in Chatbot Arena are often written purely in natural language, so the *majority* of coding-related questions in Chatbot Arena do not contain code (Chi et al., 2025); this is unlike `EDIT-Bench` and Polyglot, both of which require code for every problem. Second, there is a difference in **interaction modality.** `EDIT-Bench` and Polyglot test a model's ability to perform *instructed code edits*, where there is a freeform input (the user instruction) and structured output (the resulting code), while Chatbot Arena evaluates a model's ability to *chat*, where there is both freeform inputs and outputs. Also, the inclusion of additional code context (e.g., highlighted code) may affect correlation to Polyglot. Finally, correlation may be affected by the inclusion of **real-world user intent.** Polyglot's problems are entirely based on coding exercises from educational-style problems that lack the organic user intent present in Chatbot Arena and `EDIT-Bench`.

# 6 CONCLUSION, LIMITATIONS, AND FUTURE WORK

As instructed code edits become more widely adopted in real-world IDEs, there is a need to benchmark LLM capabilities on these types of problems. We develop a VSCode extension to collect real-world instructed code edits, which include user instructions and code contexts. We transform this in-the-wild edit data into `EDIT-Bench`, a set of high-quality test harnesses that evaluate LLM's ability to perform diverse tasks. Evaluations on 40 models show that `EDIT-Bench` is challenging even for current state-of-the-art models and provides insights into how performance varies when considering different code context information and types of edits. Overall, to adequately support developers using LLM-powered tools, our findings demonstrate the need for future models to be trained on real-world interaction modes and evaluated across a broad spectrum of problem categories, languages, code contexts, and user intents.

**Limitations and Future Work.** While we attempted to make `EDIT-Bench` as diverse as possible, there are still additions from which it would benefit. For example, as we collect more data using our extension, we will increase the number of examples we have for the existing languages and expand to other common programming languages. Additionally, despite improvements over existing benchmarks, it is unclear to what extent our problems encapsulate all real-world use cases. We plan to continue updating the `EDIT-Bench` leaderboard as new models are released and exploring automatic workflows to more seamlessly translate real-world data to benchmark problems.

## ACKNOWLEDGMENTS

This work was supported in part by the National Science Foundation grants IIS1705121, IIS1838017, IIS2046613, IIS2112471, the Department of Defense (DoD) through the National Defense Science and Engineering Graduate (NDSEG) Fellowship Program, and funding from Datadog. Any opinions, findings and conclusions or recommendations expressed in this material are those of the author(s) and do not necessarily reflect the views of any of these funding agencies.

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

# A  DATA COLLECTION DETAILS

## A.1  SYSTEM DETAILS

We adapt the prompt used in template from Continue (Continue Dev, 2025).

```
The user has requested a section of code in a file to be
    rewritten.

This is the prefix of the file:
```{language}
{prefix}
```

This is the suffix of the file:
```{language}
{suffix}
```

This is the code to rewrite:
```{language}
{code_to_edit}
```

You are an expert programmer. You will rewrite the above code to
    do the following:

{user_input}

Keep in mind indentations. Output only a code block with the
    rewritten code:
```

## A.2  GENERAL INSTRUCTIONS

Step 1: Install the extension and restart Visual Studio Code after installation. If installed successfully, you will see EditBenchExt show up on the bottom right corner of your window and the check mark changes to a spinning circle when a completion is being generated, Note, if you are using any other completion provider (e.g. Github Copilot), you must disable them when using EditBenchExt.

Step 2: EditBenchExt currently supports two main feature: read autocomplete and in-line editing (beta) below to understand how to use each one. Since we show paired responses, the way you use them are slightly different than your standard AI coding tools!

Step 3: This step is optional. If applicable, you can change what data is saved by EditBenchExt by following the instructions in "Privacy Settings".

Step 4: Create a username by clicking the EditBenchExt icon on the sidebar; detailed instructions are also in "Create an account". Your username will be used for a future leaderboard to compare individual preferences.

## A.3  PRIVACY INSTRUCTIONS

**Privacy Settings.** Your privacy is important to us. Please read carefully to determine which settings are most appropriate for you. To generate completions, the code in your current file is sent to our servers and sent to various API providers. This cannot be changed.

**Data Collection.** By default, we collect your code for research purposes. You can opt-out of this. If you are working on code containing sensitive information, we recommend that you opt out of data collection. To opt-out of data collection, please change codePrivacySettings to Debug. We will only log your code for debugging. To disable logging entirely, please change codePrivacySettings to Private. Opting-out means any bugs you encounter will be non-reproducable on our end. You can

find these settings by searching for EditBenchExt in your vscode settings or clicking the gear button of the EditBenchExt extension -> Extension Settings.

**Removing your data.** If you would like to have the option in the future for us to delete any of your data, you must create an account on EditBenchExt following instructions described in "Create an account." To remove your data, you can email any of the EditBenchExt maintainers with your username.

**Data Release.** Prior to releasing any collected code snippets to enable future research efforts, we will run a PII detector and remove any identified entities to further ensure no personal information is released.

## B EDITBENCHEXT INSTRUCTED EDITS DATA ANALYSIS

We analyze the in-the-wild data collected through EditBenchExt. We visualize the distribution of languages that users code in (Figure 6), natural languages that users write instructions in (Figure 7), length of instructions (Figure 9), length of highlighted code (Figure 8) and length of code context (Figure 10). We find that across user votes, 50.8% voted for the left response, 34.6% voted for the right response, and 14.6% voted for neither. This means that 85.4% of the time, at least one of the responses was accepted.

We also provide additional examples of user instructions across different task categories in Table 5 and the full instructions from Table 2 in Table 4. An example of highlighted user code is given in Figure 11.

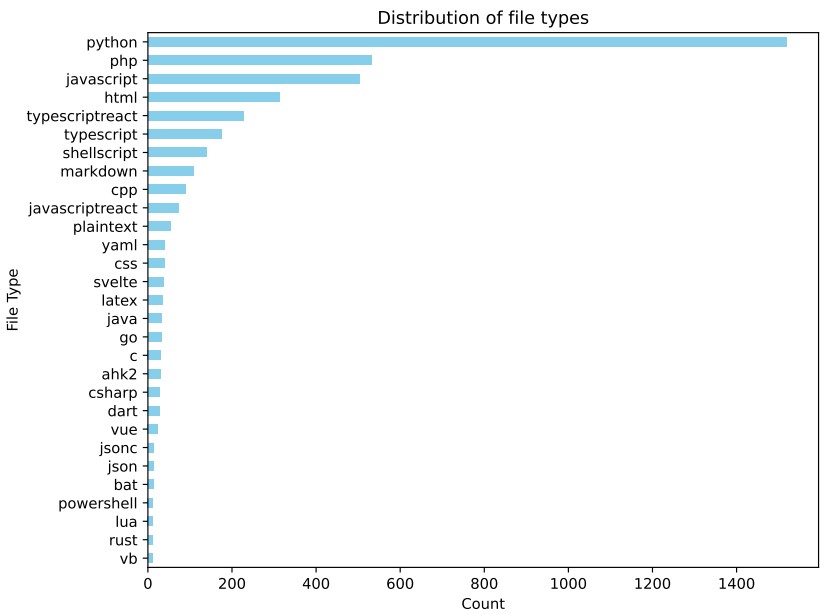

Figure 6: Distribution of file types over instructed edit users in EditBenchExt. The majority of users are working on Python code.

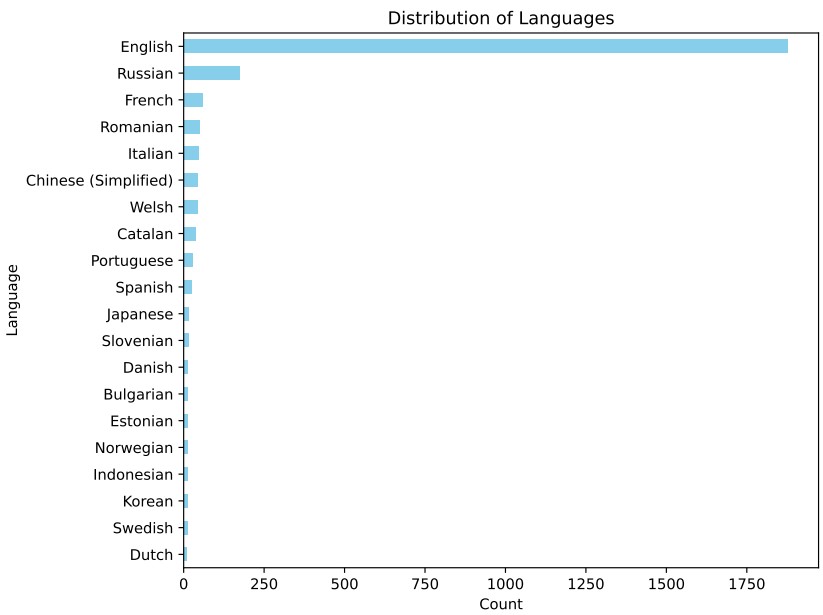

Figure 7: Distribution of natural languages in user instructions for instructed edits in EditBenchExt. The majority of users write instructions in English.

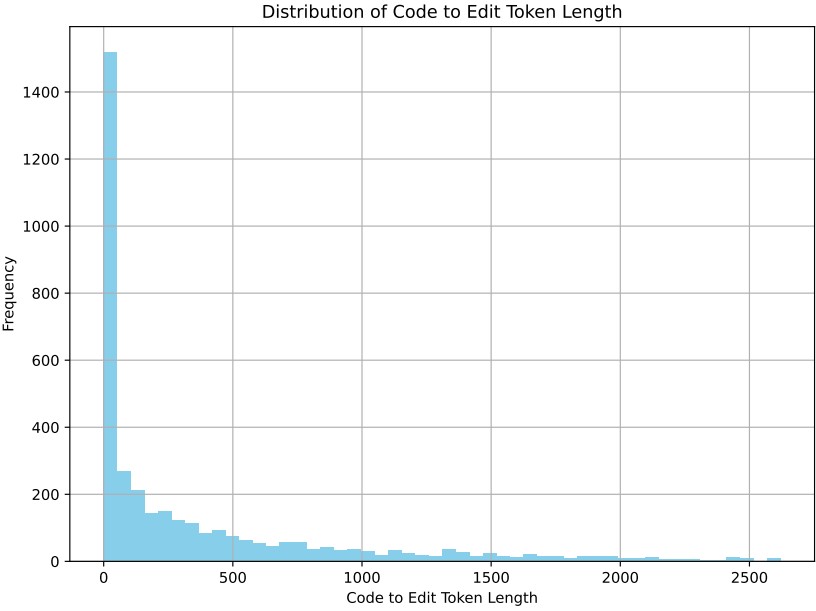

Figure 8: Distribution of highlighted code (also referred to as code to edit) token lengths. Users do not always highlight code. However, we still know their cursor placement.

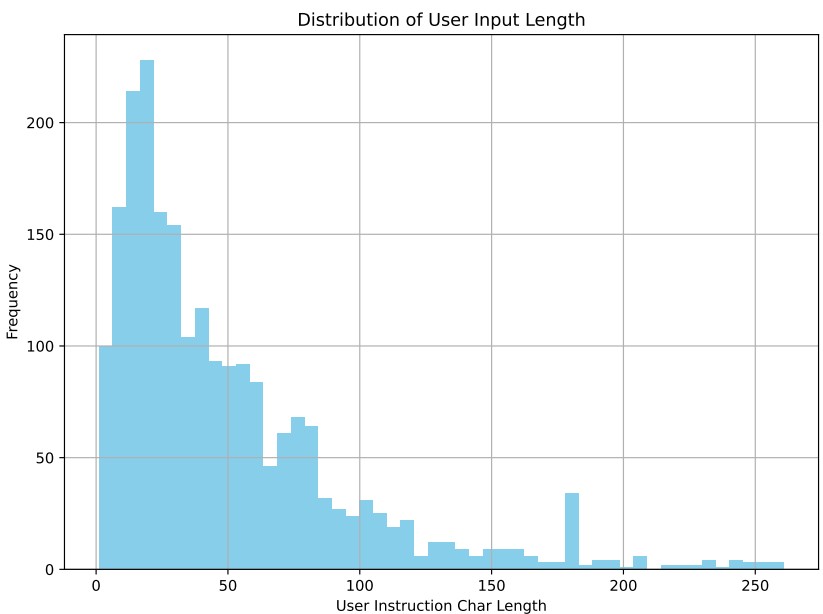

Figure 9: Distribution of the number of characters in user instructions.

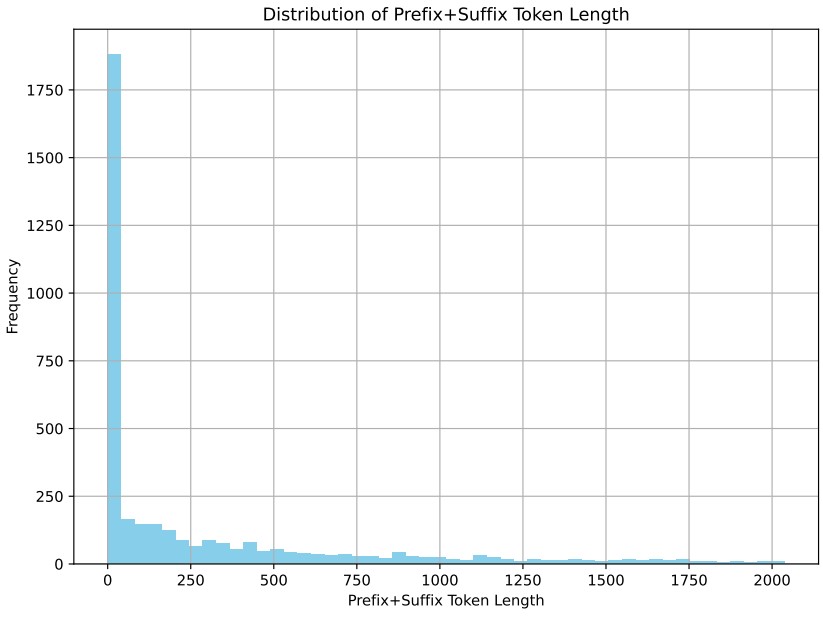

Figure 10: Distribution of context length (defined by the prefix and suffix token length) of code files.

Table 4: Full examples of user instructions across different task categories, comparing with two edit-related datasets (CanItEdit (Cassano et al., 2023b) and EditEval (Hu et al., 2023)).

| EditBench (proposed) | CanItEdit (Cassano et al., 2023b) | EditEval (Hu et al., 2023) |
|---|---|---|
| **Feature Addition** | | |
| take the globe countries layer from below ``// this'' and add it to the existing globe | Add a method 'estimate_location' that returns the estimated the appropriate location for this house, calculated by getting the average location of the top 5 most similar houses in terms of estimated price. | Add a function 'filter_odd_numbers' to filter odd numbers using lambda function. |
| **Feature Modification** | | |
| do not use R style, use python style | Flip the correlation function given to calculate the covariance instead using the Corr(X, Y), Var(X) and Var(Y). The new function should take in Corr(X, Y), Var(X) and Var(Y) in that order. | Modify the function to correctly determine the season based on month and day, considering edge cases for season changes. Raise error when invalid month is provided. |
| **Resolve Errors** | | |
| RuntimeError: Cannot close a running event loop sys:1: RuntimeWarning: coroutine 'Application.shutdown' was never awaited sys:1: RuntimeWarning: coroutine 'Application.initialize' was never awaited | Fix combination_unlimited _rep() so that it returns the right result. The function combination _unlimited_rep should be returning the combination of n-r+1 and n by calling on combination() with those arguments. | Fix the given function to correctly identify whether a string represents a valid floating-point number or not, including handling edge cases such as scientific notation (e.g., '1e-4'), positive and negative signs, and leading/trailing whitespace. Ensure the function is robust and handles exceptions appropriately. |
| **Optimize Code** | | |
| optimize the computation by better batching the latter part | Optimize the bm25 algorithm by avoiding frequency calculations. | Optimize the function to find the longest common subsequence for the given two sequences using dynamic programming |

Table 5: Additional examples of user instructions across different task categories, comparing with two edit-related datasets (CanItEdit (Cassano et al., 2023b) and EditEval (Hu et al., 2023)).

| EditBench (proposed) | CanItEdit (Cassano et al., 2023b) | EditEval (Hu et al., 2023) |
|---|---|---|
| **Feature Addition** | | |
| `add example usage` | `Add a method called 'header' which returns the header of a csv file as a list.` | `Add a check for None to prevent possible null reference exceptions in the 'editorial_reviews' function.` |
| **Feature Modification** | | |
| `modify the cmap so the displayed values are the same as the text displayed on the raw map.` | `Modify the 'Quiz' class to allow the user to skip a question using 'self.skip_question()', and record the number of questions that were skipped in 'self.skipped'.` | `Modify the function to return the word with the most number of occurrences in the given list of strings. If there are multiple words with the same maximum occurrences, return all of them in a list sorted alphabetically.` |
| **Resolve Errors** | | |
| `theta -= alpha * gradient ValueError: non-broadcastable output operand with shape (2,1) doesn't match the broadcast shape (2,3)` | `Fix the methods in 'Course' so that they never throw errors. Even when 'len(self.students) == 0'. Instead they should return 'None'. Additionally, do not use the words 'for', 'while', or 'map' anywhere in the code. You should accomplish this using higher order functions.` | `Fix the function to correctly find the single element in a sorted array where every other element appears exactly twice.` |
| **Optimize Code** | | |
| `run these in parallel` | `Optimize the AI to find the best move in less steps.` | `Optimize the given function to find the first position of an element in a sorted array.` |

**Example EditBench original_code.py**

```python
from langchain_openai import ChatOpenAI
from langchain.prompts import PromptTemplate
from langchain.chains import LLMChain
from langchain_community.retrievers import BM25Retriever
from os import getenv
# omit some imports for spacing
load_dotenv()
st.title("CardioRAG")
# load in PDF for RAG
if "retriever" not in st.session_state:
    st.text("Loading PDF...")
    prog_bar = st.progress(0)
    pdf_reader = PyPDF2.PdfReader(open("Moss and Adams 10e Vol 1 & 2.pdf", 'rb'))
    chunks = []
    for page_num in range(60, 600):
        prog_bar.progress((page_num-60)/(600-60))
        chunks.append(pdf_reader.pages[page_num].extract_text())
    # put chunks into vector store
    retriever = BM25Retriever.from_texts(chunks, metadatas=["page_num": p  for p in
↪   range(60, 600)], preprocess_func=word_tokenize)
    st.session_state["retriever"] = retriever
st.text("Loaded PDF")
if "messages" not in st.session_state:
    st.session_state["messages"] = [

        "role": "assistant", "content": "Hi, I'm a chatbot who has read the Moss &
↪   Adams Cardiology textbook. How can I help you?"
    ]

with st.form("chat_input", clear_on_submit=True):
    a,b = st.columns([4,1])
    user_input = a.text_input(
        label="Question:",
        placeholder="What is the incidence of congenital heart disease?",
        label_visibility="collapsed",
    )
    b.form_submit_button("Send", use_container_width=True)
for i, msg in enumerate(st.session_state.messages):
    message(msg["content"], is_user=msg["role"] == "user", key=str(i))
if user_input and st.session_state["password"]:
    st.session_state.messages.append("role": "user", "content": user_input)
    message(user_input, is_user=True, key=str(len(st.session_state.messages) - 1))
    llm = ChatOpenAI(
        api_key=getenv("OPENROUTER_API_KEY"),
        base_url="https://openrouter.ai/api/v1",
        model_name="meta-llama/llama-3.2-3b-instruct",
        streaming=True)
    retriever = st.session_state["retriever"]
    docs = retriever.get_relevant_documents(user_input)
    DIVIDER = "-"*10
    context = DIVIDER.join([f"Page d.metadata['page_num']: d.page_content" for d in
↪   docs])

    prompt = PromptTemplate(
        input_variables=["context", "question"],
        template="""You are a helpful AI assistant who has read the Moss & Adams
↪   Cardiology textbook.  Use the following context to answer the question. If you
↪   don't know the answer, just say you don't know.
Context: context
Question: question
Answer:"""
    )
    print(prompt)
    chain = LLMChain(llm=llm, prompt=prompt)
    response = chain.run(context=context, question=user_input)
    st.session_state['messages'].append("role": "assistant", "content": response)
    message(response, key=str(len(st.session_state.messages) - 1))
```

Figure 11: Example code file and highlighted section. The user instruction for this file: "Can you edit this to work with streaming responses?"

## C  EDITBENCH DETAILS

**Data Curation and Programming Languages.** We started with 999 Python and 234 Javascript files. We curated (Phase 1) down to 370 Python and 100 Javascript files. We then successfully tested and annotated 104 Python and 9 JavaScript problems. React represents its own ecosystem in Javascript; 5 out of 9 of our problems are based on React.

**Library Distribution.** EditBench contains 74 unique imports for Python (Figure 3). We also calculated distributions for CanItEdit (25 unique imports), Polyglot (15 unique imports), and EditEval (16 unique imports) (Figure 12).

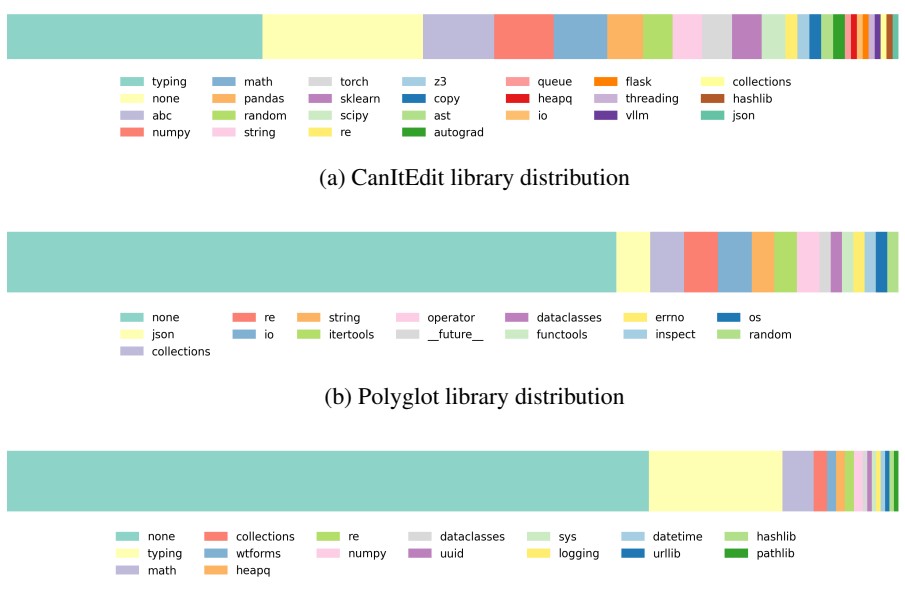

(a) CanItEdit library distribution

(b) Polyglot library distribution

(c) EditEval library distribution

Figure 12: Library distributions for comparison benchmarks: CanItEdit (25 unique imports), Polyglot (15 unique imports), and EditEval (16 unique imports).

**Problem Filtering.**  When filtering in-the-wild data, we discarded problems that were too easy and too ambiguous. Examples of problems that are too easy:

- Given the instruction "decrease the speed" and the highlighted code snippet that clearly includes `self.master.after(30, self.game_loop)  Adjust speed here (milliseconds)`. It is obvious that the change is trivial, as it just involves increasing the value of the hard-coded value.
- Given the instruction "add api key" and the highlighted code snippet `chat_model = ChatOllama(model="llama3.2", base_url="http://localhost:11434")`. It is clear that the change would simply involve adding an api key parameter.

Examples of problems that are too ambiguous:

- Given the instruction "find and solve problems" and code context consisting of dozens of lines of Python code to instantiate an ML training pipeline with no obvious issue. From the annotator's perspective, it is unclear what problem the user was intending the LLM to fix.
- Given the instruction "The code does not seem to implement all the logic please extend it to make all logic work." and a short highlighted snippet (e.g., `list_available_resolutions(yt)`). From the annotator's perspective, it is unclear what "logic" needs to be implemented and the highlighted code provides insufficient context.

Table 6: Each model in our experiments with their official names and provider links

| Model | Model Size | Proprietary | Link to Provider |
|---|---|---|---|
| gpt-4o-mini | Unknown | True | https://platform.openai.com/docs/models/gpt-4o-mini |
| gpt-4o | Unknown | True | https://platform.openai.com/docs/models/gpt-4o |
| gpt-5-nano | Unknown | True | https://platform.openai.com/docs/models/gpt-5-nano |
| gpt-5-mini | Unknown | True | https://platform.openai.com/docs/models/gpt-5-mini |
| gpt-5 | Unknown | True | https://platform.openai.com/docs/models/gpt-5 |
| gpt-o3-mini | Unknown | True | https://platform.openai.com/docs/models/o3-mini |
| gpt-o4-mini | Unknown | True | https://platform.openai.com/docs/models/gpt-4o-mini |
| gpt-oss-20b | 20b | False | https://platform.openai.com/docs/models/gpt-oss-20b |
| gpt-oss-120b | 120b | False | https://platform.openai.com/docs/models/gpt-oss-120b |
| sonnet-3.5 | Unknown | True | https://docs.anthropic.com/en/docs/about-claude/models/overview |
| sonnet-3.7 | Unknown | True | https://docs.anthropic.com/en/docs/about-claude/models/overview |
| sonnet-4 | Unknown | True | https://docs.anthropic.com/en/docs/about-claude/models/overview |
| glm-4.5 | 355b | False | https://openrouter.ai/z-ai/glm-4.5 |
| gemma-3n-e4b-it | 8b | False | https://openrouter.ai/google/gemma-3n-e4b-it |
| gemma-3-12b-it | 12b | False | https://openrouter.ai/google/gemma-3-12b-it |
| gemma-3-27b-it | 27b | False | https://openrouter.ai/google/gemma-3-27b-it |
| gemini-2.5-flash | Unknown | True | https://openrouter.ai/google/gemini-2.5-flash |
| gemini-2.5-pro | Unknown | True | https://openrouter.ai/google/gemini-2.5-pro |
| grok-4-fast | Unknown | True | https://openrouter.ai/x-ai/grok-4-fast:free |
| grok-code-fast-1 | Unknown | True | https://openrouter.ai/x-ai/grok-code-fast-1 |
| kimi-k2 | 1T | False | https://openrouter.ai/moonshotai/kimi-k2-0905 |
| qwen-2.5-coder-32b-instruct | 32B | False | https://openrouter.ai/qwen/qwen-2.5-coder-32b-instruct |
| qwen-2.5-coder-72b-instruct | 72B | False | https://openrouter.ai/qwen/qwen-2.5-72b-instruct |
| qwen-3-4b | 4B | False | https://openrouter.ai/qwen/qwen3-4b:free |
| qwen-3-8b | 8B | False | https://openrouter.ai/qwen/qwen3-8b |
| qwen-3-14b | 14B | False | https://openrouter.ai/qwen/qwen3-14b |
| qwen-3-30b-a3b | 30B | False | https://openrouter.ai/qwen/qwen3-30b-a3b |
| qwen-3-coder-flash | Unknown | True | https://openrouter.ai/qwen/qwen3-coder-flash |
| qwen-3-coder | 405B | False | https://openrouter.ai/qwen/qwen3-coder |
| deepseek-v3-chat | 671B | False | https://openrouter.ai/deepseek/deepseek-chat-v3.1 |
| deepseek-r1 | Unknown | False | https://openrouter.ai/deepseek/deepseek-r1-0528 |
| llama-4-maverick | Unknown | False | https://openrouter.ai/meta-llama/llama-4-maverick |
| llama-4-scout | Unknown | False | https://openrouter.ai/meta-llama/llama-4-scout |
| llama-3.1-405B | 405B | False | https://openrouter.ai/meta-llama/llama-3.1-405b |
| llama-3.3-70B | 70B | False | https://openrouter.ai/meta-llama/llama-3.3-70b-instruct |
| llama-3.3-8b | 8B | False | https://openrouter.ai/meta-llama/llama-3.3-8b-instruct:free |
| mistralai-devstral-small | 24B | False | https://openrouter.ai/mistralai/devstral-small |
| mistralai-devstral-medium | Unknown | True | https://openrouter.ai/mistralai/devstral-medium |
| mistralai-codestral-2508 | Unknown | True | https://openrouter.ai/mistralai/codestral-2508 |
| mistral-small-3.2-24b-instruct | 24b | False | https://openrouter.ai/mistralai/mistral-small-3.2-24b-instruct |

## D   EVALUATION SET-UP

**Prompts.** We evaluated models using three prompting strategies. The `Whole` (i.e., +Highlight) prompt tasks the model with regenerating the entire code file (Figure 13). The `Cursor Position` (i.e., +Highlight, +Cursor) prompt is the same as the `Whole` prompt with the user's cursor position added (Figure 14) (i.e., Code Only). The `No Highlight` prompt is the same as the `Whole` prompt but information about user's highlights are removed (Figure 15) Note that the prompt in Section A is slightly different due to the settings (e.g., cost, response time, etc.)

**Model Access** We used the OpenAI API to query the GPT models, the Anthropic API for the Claude models, and OpenRouter for access to all other models. The full list of official model names and links to the providers is in Table 6.

**Model Parameters.** For every model provider, the default settings were used. The `gpt-o3-mini`, `gpt-o4-mini`, and `gpt-5` models used the default medium effort for reasoning while `gpt-o3-mini (high)`, `gpt-o4-mini (high)`, and `gpt-5 (high)` used the high reasoning effort setting.

**Evaluation Environment.** To isolate our testing environment, we ran all our evaluations inside of a Docker container. We used the Ubuntu 22.04 image for our container. The Dockerfile for building our container will be provided with the release of our benchmark.

Generate a new implementation of the following code based on the user instruction:

The Original code (to be modified):

```{lang}
{original_code}
```

The user instruction is:
{instruction}

And they highlighted this section to be changed:
```{lang}
{highlighted_code}
```

Please only change the highlighted section and leave the rest of the code unchanged.
Please output the entire code file.
Respond only in a code block beginning with ```{lang}.

Figure 13: `Whole` prompt given to models

Generate a new implementation of the following code based on the user instruction:

The Original code (to be modified):

```{lang}
{original_code}
```

The user's cursor position (line number: column number) is at {cursor_pos}

The user instruction is:
{instruction}

And they highlighted this section to be changed:
```{lang}
{highlighted_code}
```
Please only change the highlighted section and leave the rest of the code unchanged.
Please output the entire code file.
Respond only in a code block beginning with ```{lang}.

Figure 14: `Cursor Position` prompt given to models

Generate a new implementation of the following code based on the user instruction:

The Original code (to be modified):

```{lang}
{original_code}
```

The user instruction is:
{instruction}

Please output the entire code file.
Respond only in a code block beginning with ```{lang}.

Figure 15: `No Highlight` prompt given to models

# E   ADDITIONAL EVALUATION RESULTS

**Effect of context length.**   We also conduct additional analysis by binning performance into short, medium, and long. Perhaps unsurprisingly, we see that models tend to do better on shorter context length problems. In general, the worse a model is overall, we also see that it has a much larger gap between the best and worst bin (e.g., `gemma-3n-e4b-it` has a 34.2% gap and `gpt-oss-120b` a 33.6% gap).

Table 7: Effect of context length on average pass@1.

| Context Bin | Average Pass@1 |
|---|---|
| Short (i.e., < 1k chars) | $71.03 \pm 7.60$ |
| Medium (i.e., 1k–3k chars) | $62.09 \pm 8.56$ |
| Long (i.e., > 3k chars) | $59.94 \pm 10.43$ |

**Instruction and Highlight Length analysis.**   Given the large gap between `easy` and `hard` problems, we explore what types of prompts are present in `hard` problems compared to the general dataset. As shown below, we see that `hard` instructions tend to have *shorter* instructions (by nearly 5 times) but slightly *longer* highlighted code.

Table 8: Comparing instruction and highlight length for easy versus hard questions.

| | Instruction Length (chars) | Highlight Length (chars) |
|---|---|---|
| Easy Questions | $351.21 \pm 1018.87$ | $942.30 \pm 1275.35$ |
| Hard Questions | $75.09 \pm 107.20$ | $881.45 \pm 1275.23$ |

**Code Context Dependent Example.**   Additional code context is often crucial to understanding and solving a problem. This can be because the code context is simply too long or because the user instruction is too ambiguous. Let us take problem 45 in `EDIT-Bench` as an example.

In this example, the user instruction is to 'remove', which could mean the removal of the class, the function, or the implementation of the remove functions. However, when observing the problem we can consider the following from the rest of the code context:

1. There is no highlighted code segment. This means it is impossible for the user intent to be removal as the only available operation is to add code.

2. The remove_vertex and remove_edge functions appear multiple times in the code. However, the function implementations are implemented incorrectly in the original code.

Thus, the other interpretations of 'remove' make little sense given the entire context of the problem. The correct answer can be inferred from the rest of the context, but would be difficult to understand from the instruction alone.

## F  FURTHER DISCUSSION

### F.1  LIMITATIONS

In addition to the limitations in Section 6, we discuss several more below:

**Limited Programming and Natural Languages.** Although EditBench contains problems in both Python and Javascript as well as several non-English languages, the amount of Javascript is limited. We aim to continue collecting more data and building test harnesses and problems for more programming and natural languages.

**Contamination.** One major challenge with releasing benchmarks is that future models may accidentally (or intentionally) be trained on the benchmark itself. We have taken pre-emptive measures to prevent this by ensuring the dataset documentation contains instructions to prevent any accidental scraping of our data. Following recent benchmarking efforts (White et al., 2024; Jain et al., 2024), we will also aim to make our pipeline more automatic. Combined with the continuous stream of data from EditBenchExt, new problems can be continuously released, preventing data contamination. We discuss this in more detail in Section F.2.

### F.2  FUTURE WORK

In addition to increasing the number of examples for the existing languages and expanding to other common programming languages, we plan to continue updating the `EDIT-Bench` leaderboard as new models are released.

**Automatic Test Harness Generation** When we evaluated our fully-agentic pipeline on our model generations, all models achieved a `pass@1` of 0%. This indicates that these test cases were either broken or too constrained to be usable in the benchmark. Given that prior research indicates models are at least somewhat capable of generating well-specified test cases (Mündler et al., 2025), we suspect that models are still unable to fully understand the intent behind in-the-wild user instructions. Given that we have a continuous stream of data from EditBenchExt, resolving this will be key to enabling fully automatic test harness generation for EditBench. In general, we also believe that improving an agent's ability to generate test harnesses constitutes an interesting avenue for future research.

### F.3  BROADER IMPACT.

This paper presents work whose goal is to advance the field of Machine Learning. Due to the ethical and user privacy considerations involved with storing and releasing user code data, we take a conservative approach to data release. Despite giving users full control over their privacy, we have at least two annotators who provide additional screening for Personally Identifiable Information (PII) on each problem during our data curation and release process. We will continue to screen for PII as we release more problems.

