# OpenReview forum: "EDIT-Bench: Evaluating LLM Abilities to Perform Real-World Instructed Code Edits"
_ICLR.cc/2026/Conference — ICLR 2026 Oral_

### Official Review · Reviewer_SuE5 · 2025-10-30

**Soundness:** 3
**Presentation:** 2
**Contribution:** 3
**Rating:** 6
**Confidence:** 3

**Summary:**

The paper presents EditBench, a benchmark designed to evaluate large language models (LLMs) on real-world instructed code editing, where LLMs are guided by natural language to modify existing code. To address the lack of realistic interactions, the authors built a VS Code extension that collects in-the-wild user instructions, highlighted code, and cursor positions from nearly 500 developers. After filtering and manual annotation, they curated 545 problems across five natural languages (English, Spanish, Russian, Chinese, Portuguese) and two programming languages (Python, JavaScript), each paired with test harnesses for automated evaluation. Evaluating on 40 LLMs show that EditBench is highly challenging where only 3 models (Claude-Sonnet-4, GPT-o3-mini-high, Claude-3.5-Sonnet) exceed 60% pass@1. Including highlighted code improves performance by up to 7%, while cursor position lowers it.

**Strengths:**

1. EditBench is collected directly from real-world interactions, which makes it more representative of practical coding workflows.
2. EditBench incorporates diverse user instructions, code context, highlighted code, and cursor position.
3. Multilingual benchmark across five natural and two programming languages broadens applicability.
4. Comprehensive evaluation with 40 LLMs spanning reasoning and non-reasoning models.

**Weaknesses:**

1. The authors keep stating that they include "both interesting and challenging (line 158)" or removing "ambiguous (line 178)" problems, but there is a lack of explanation what problem refers to bad or good problem.
2. In the evaluation, the authors split EditBench into easy and hard groups, but they do not provide clear criteria explaining how this division was made.
3. Following up on the previous concern, the authors claim that “\emph{hard instructions tend to have shorter instructions (by nearly 5 times) but slightly longer highlighted code}.” It is unclear whether the shorter instructions are compared to those in easy problems or to the length of the highlighted code itself. For either perspective, there is a lack of in-depth analysis to figure out the root cause.
4. From the example shown in Table 5, the instructions collected by EditBench appear less clear than the annotator-provided prompts in other benchmarks. This may explain why all LLMs perform poorly with such instructions. It would be helpful to know what the upper bound of performance is when following these instructions.
5. It is notable that older reasoning models such as o3 and o3-mini-high outperform GPT-5. A deeper analysis of this result would be valuable to understand why newer models lag behind in this setting.

**Questions:**

1. What specific criteria were used to determine which problems were considered “interesting and challenging”?
2. How did the authors define and measure problem difficulty when splitting EditBench into easy and hard groups?
3. What is the estimated upper bound of model performance when following the instructions in EditBench, and was any human or expert-based evaluation used to assess the clarity and followability of these instructions?

---

> ### Author Response · Authors · 2025-11-21
>
> Thank you for the helpful comments and appreciation of our work. We aim to address your comments below:
>
> **[Weakness 1 / Question 1: Problem filtering]** When filtering in-the-wild data, we discarded problems that were too easy and too ambiguous. An example of a “too easy” problem is the following: given the instruction *“decrease the speed”* and the highlighted code snippet:
> ```python
>  def game_loop(self):
>         self.move_ball()
>         self.master.after(30, self.game_loop)  # Adjust speed here (milliseconds)
> ```
> The change is trivial, as it just involves increasing the value of the hard-coded value. An example of a “too ambiguous” problem is the following: given the instruction *“find and solve problems”* and code context consisting of dozens of lines of Python code to instantiate an ML training pipeline with no obvious issue. From the annotator's perspective, it is unclear what problem the user was intending the LLM to fix. We include these examples and more in the revised pdf to increase transparency of our problem filtering process.
>
> **[Weakness 2 / Question 2: Easy/hard split]** Following Gauthier et al, “we categorize problems that were solved by k or fewer models as Hard and the remainder as Easy”, as mentioned in L313-316 of the submission.
>
> **[Weakness 3: Instruction vs. highlight analysis]** To clarify “whether the shorter instructions are compared to those in easy problems or to the length of the highlighted code itself,” we provide a table to break down exactly the difference between instruction and highlight lengths in easy versus hard problems.
> |                   | Instruction Length (chars)     | Highlight Length (chars)     |
> |-------------------|--------------------------------|-------------------------------|
> | Easy Questions    | 351.21 +/- 1018.87             | 942.30 +/- 1275.35           |
> | Hard Questions    | 75.09 +/- 107.20               | 881.45 +/- 1275.23           |
>
> **[Weakness 4 / Question 3: Upper bound on performance]** We actually found that approximately 95% of problems were solved by at least one model; thus, the lowest upper bound on performance is 95%.
>
> **[Weakness 5: Why newer reasoning models lag behind]** As we discuss in L366 of the submission, “when inspecting test cases where gpt-5 failed, we find that it struggles with simple tasks like formatting code indentation properly and catching edge cases, despite being a strong reasoning model.” We confirm this by checking gpt-5 performance on hard versus easy problems and see that, in particular, gpt-5 performs worse than o3-mini on easy problems (e.g., 86.0% vs 91.9%).

---

> > ### Comment · Reviewer_SuE5 · 2025-11-26
> > **Reviewer Response**
> >
> > Thanks for the clarification. My questions have been resolved. I suggest that the author add those details to their revisions for better presentation.

---

### Official Review · Reviewer_aKBn · 2025-10-31

**Soundness:** 4
**Presentation:** 4
**Contribution:** 4
**Rating:** 10
**Confidence:** 5

**Summary:**

The paper presents a benchmark for code editing, where code editing
means working with a VSCode extension and using a sidebar LLM chat window
to edit code. The paper recruited 500+ participants to use a VSCode extension
that was developed by the authors. They received  appropriate IRB
approval and take care to manage PII.

The paper then puts in the effort to turn the raw data into a benchmark
suite with test cases. The benchmark is quite challenging and much more
realistic than prior work in this space.

**Strengths:**

I think this paper sets a new bar for coding benchmarks. Almost every other
benchmark that I can think of for any coding tasks (and not just code editing)
uses artificial tasks and prompts. The only related work I can think of that
uses natural prompts is the StudentEval benchmark (Babe et al, 2023), but the
tasks in that are quite trivial compared to what is presented here.

I have gotten quite tired of evaluating LLMs on made up tasks that I know are
trivial compared to the kinds of prompts that get used in the wild. This paper
helps change that. I would personally use this benchmark in my own work. It
is easy to see that other researchers would also find it very useful.

**Weaknesses:**

- The only weakness I can think of is that the context provided to the
  models seems to be just the current editor window. Tools like Cursor
  are well beyond this and supply significantly more context. However,
  I think this simplification is the right one for a benchmark.

**Questions:**

- I don't quite understand Appendix A.1. This looks like the system prompt
  for editing. What does "collecting user votes" means? Actually, section D
  makes it clear that this is not the prompt template for the model, so I
  am quite confused.

---

> ### Author Response · Authors · 2025-11-21
>
> Thank you for the helpful comments, and we are excited to see the interest in using EditBench for your own work! We aim to address your comments below:
>
> **[Weakness 1: Increase context provided]** We agree that future iterations of EditBench can consider evaluating LLMs with multiple files as context and have added this as an interesting direction for future work in the revised submission. We note that doing so would also require controlling for the retrieval method that the LLM uses, since many repositories contain more files than can naturally fit most models’ context windows.
>
> **[Question 1: Appendix clarifications]**
> - “Collecting user votes” was a misprint on our end and has been removed.
> - Appendix A.1 was the prompt used in the wild, and Section D was the prompt used during our experiments. The slight differences in the prompt are to account for the difference in settings (e.g., cost, response time, etc.). We’ve added this clarification in section D.

---

> > ### Comment · Reviewer_aKBn · 2025-11-27
> >
> > I have this this response and will keep the same score.

---

### Official Review · Reviewer_CbSq · 2025-10-31

**Soundness:** 2
**Presentation:** 3
**Contribution:** 3
**Rating:** 8
**Confidence:** 3

**Summary:**

This paper introduces a new benchmark, EditBench, for evaluating LLM code editing capabilities grounded in real-world usage, i.e., user instructions and code contexts collected in the wild. EditBench comprises of 545 problems, multiple natural and programming languages, and a diverse set of real-world use cases, ranging from resolving errors to adding features. 40 diverse LLMs have been evaluated and only 3 models score over 60% on EditBench, indicating it is a challenge set of problems. Also, empirical results and analyses show that, varying levels of contextual information greatly affect task success rate.

**Strengths:**

1. This paper proposes a new benchmark EditBench for evaluating LLM code editing capabilities grounded in real-world usage.
2. Only 3 models score over 60\% on EditBench, indicating it is a challenge set of problems.
3. The presentation is good and the paper is easy to follow.

**Weaknesses:**

1. No codebase is released or provided to confirm the reproducibility.

**Questions:**

See Weaknesses.

---

> ### Author Response · Authors · 2025-11-21
>
> Thank you for the helpful comments and appreciation of our work. We aim to address your comment below:
>
> **[Weakness 1: Code release]** In the supplementary materials, we originally included a jsonl file containing all problems in EditBench, along with test cases. We have updated the supplementary materials to include a full code repository, which makes it easy for anyone to generate edits with new models with the exact same prompts used in our work and evaluate the outputs using the same metrics.

---

### Official Review · Reviewer_yRbL · 2025-11-02

**Soundness:** 3
**Presentation:** 4
**Contribution:** 3
**Rating:** 6
**Confidence:** 4

**Summary:**

This paper introduces EditBench, a novel benchmark designed to evaluate Large Language Models' (LLMs) capabilities in instructed code editing. Unlike existing benchmarks that rely on synthetic or competitive programming problems, EditBench's key contribution is its data, which is sourced from real-world developer workflows. Data was collected via a custom VS Code extension from nearly 500 users, capturing user instructions and code context in the wild. The benchmark comprises 545 problems spanning diverse real-world use cases, from bug fixes to feature additions. EditBench is the first to systematically require models to leverage rich contextual cues—including the user instruction, the entire code file, highlighted code snippets, and cursor position—to solve problems, thus more accurately simulating a developer's environment. It features diversity across 5 natural languages (including Chinese, English, etc.) and Python/Javascript programming languages. The authors evaluated 40 different LLMs, finding that EditBench is highly challenging; even the best-performing model (claude-sonnet-4) achieved only 66.67% pass@1. The study further demonstrates the significant impact of contextual information and task categories on model performance.

**Strengths:**

S1: The primary strength is the in-the-wild data collection methodology, which captures non-templated, natural user instructions and context from a real IDE environment, reflecting authentic developer challenges.

S2: The use of a rigorous process involving human experts and secondary review to create the test suite addresses the inherent difficulty of creating a verifiable benchmark from raw user behavioral data, ensuring the high quality and trustworthiness of the results.

S3: The design explicitly tests the models' ability to integrate contextual cues (file content, highlights, cursor), which is shown to be crucial for performance, thus providing valuable insights for future LLM development.

**Weaknesses:**

W1: EditBench currently supports only Python and Javascript. For a benchmark aiming for real-world representation, the lack of support for other major industry languages like Java, C++, Go, C#, and TypeScript limits its broad applicability.

W2: The extended version, EditBench-complete, relies on GPT-4o for translating non-English instructions and comments. While this increases linguistic diversity, the paper lacks detailed evidence of human quality validation for the LLM-translated content. This raises a concern about potential loss of naturalness or subtle nuance of user intent in non-English problems, potentially diminishing the "in-the-wild" authenticity of this subset of data.

W3: The paper mentions code context lengths can be long (average file length 4.5k tokens) but provides minimal dedicated analysis on model performance differences concerning context length. Given that LLM performance on long-context tasks is a major research area, a specialized analysis contrasting performance across different context length bins (e.g., short, medium, long) would have been highly valuable.

**Questions:**

Q1: The paper notes that models perform best when given highlighted code but not the cursor position, and that context can affect performance by up to 11%. Could the authors provide a more detailed ablation study in the appendix or rebuttal, clearly listing performance metrics for all four combinations: (a) No extra context; (b) Highlighted code only; (c) Cursor position only; (d) Highlighted code and cursor position. This would clarify why the cursor position appears to have a detrimental or unhelpful effect and quantify the exact utility of each cue.

Q2: Given that non-English problems in EditBench-complete were generated via GPT-4o translation, what specific quality control steps were taken to verify the naturalness and preservation of the original user intent in these translations? Were the non-English problems (e.g., Chinese, Russian, etc.) reviewed by native speakers to ensure quality?

---

> ### Author Response · Authors · 2025-11-21
>
> Thank you for the helpful comments and appreciation of our work. We aim to address your comments below:
>
> **[Weakness 1: Support more languages]** We agree that those languages would increase the broad applicability of EditBench. We have also collected data for C++, TypeScript, and Java, and are currently gauging interest in expanding the benchmark to cover these languages.
>
> **[Weakness 2 / Question 2: Validating translations]** We had native speakers evaluate a subset of the translated tasks, primarily in Chinese and Spanish. We experimented with several models (GPT-4o, GPT-4o-nano, GPT-4o-mini) and Google Translate. We found GPT-4o to provide the best quality with no noticeable concerns with any of the translations.
>
> **[Weakness 3: Differences across context lengths]** We have added an analysis based on the reviewer’s suggestion by bucketing our problems into short/medium/long context bins. Perhaps unsurprisingly, we see that models tend to do better on shorter context length problems. In general, the worse a model is overall, we also see that it has a much larger gap between the best and worst bin (e.g., gemma-3n-e4b-it has a 34.2% gap and gpt-oss-120b a 33.6% gap). We have added this analysis to Appendix E of the revised submission.
> |                         | Average Pass@1    |
> |-------------------------|-------------------|
> | Short (i.e., < 1k chars)   | 71.03 +/- 7.60    |
> | Medium (i.e., 1k–3k chars) | 62.09 +/- 8.56    |
> | Long (i.e., > 3k chars)    | 59.94 +/- 10.43   |
>
> **[Question 1: Cursor position ablation]**  We provide the experimental results for all four combinations. From the table below, we see that “Cursor Only” is not as useful for models as “Highlight Only”, though both tend to be individually more useful than the combination.
> | Model Name | No Extra Context | Highlight Only | Cursor Only | Highlight and Cursor |
> |------------|-----------|------------|-------------|--------------------|
> | claude-sonnet-4 | 60.19 | 66.67 | 62.96 | 64.81 |
> | gpt-o3-mini | 56.48 | 63.89 | 59.26 | 52.78 |
> | gemini-2.5-pro | 49.53 | 55.66 | 53.70 | 55.56 |
> | deepseek-chat-v3.1 | 53.70 | 58.88 | 57.41 | 51.85 |
> | qwen3-coder-flash | 55.14 | 56.48 | 54.63 | 50.93 |

---

### Author Response · Authors · 2025-12-03
**Summary of Response and Discussion**

We thank all reviewers for their positive and constructive feedback. We appreciate that the reviews recognize EditBench’s contribution toward evaluating LLM coding capabilities on “non-templated, natural user instructions and context from a real IDE environment” (yRbL) that are “representative of practical coding workflows” (SuE5), which constitute a “a challenging set of problems” (CbSq). We are especially excited to see that Reviewer aKBn “would personally use this benchmark in my own work.”

In our rebuttal, we believe we have strengthened the final version and addressed remaining concerns.
- We provided details on how we performed human validation of translations across languages (yRbL), along with how we filtered out problems that were too easy or ambiguous (SuE5).
- We conducted additional analysis on context length ranges across the problem set (yRbL).
- We clarify instruction vs highlight length for problem difficulty via an additional table (SuE5).
- We additionally ablated the effect of cursor position in isolation (yRbL).
- We clarified the upper bound performance on the benchmark (SuE5).
- We updated supplementary code that allows future work to replicate our analyses and easily evaluate new models on EditBench (CbSq).

All of the additional details and results have already been added to the revised PDF. We note that reviewer replies indicate that our replies satisfied their questions and concerns.

---

### Meta-Review · Area_Chair_gFQb · 2026-01-06

**Summary:**

Reviewers are generally positive about this submission. Many concerns are about extending the scope of the work, such as expanding it to cover more programming languages or to enable a longer programming context.
Otherwise, there are a few concerns about the lack of clarity in the paper writing, the need for further analysis or ablation studies, and the concern about the machine-translated multilingual annotations of the dataset.

**Reviewer Concerns:**

All critical concerns about the dataset quality, clarification, and missing analysis and ablation studies are well addressed.
For concerns about extending the scope of the work, the authors describe them as future work, which makes sense to me.

**Reviewer Scores:**

One reviewer (aKBn) replied that they will keep the same score (10). Another reviewer (SuE5) responded with satisfaction. For the other two reviewers, they both gave positive scores and will likely keep it unchanged or even increase it.

---

### Decision · Program_Chairs · 2026-01-26

Accept (Oral)